# Juxtacellular opto-tagging of hippocampal CA1 neurons in freely moving mice

Lingjun Ding[1,2,3], Giuseppe Balsamo[1,2,3], Hongbiao Chen[1,2,3], Eduardo Blanco-Hernandez[1,2], Ioannis S Zouridis[1,2,3], Robert Naumann[4,5], Patricia Preston-Ferrer[1,2]*, Andrea Burgalossi[1,2]*

[1]Institute of Neurobiology, Eberhard Karls University of Tübingen, Tübingen, Germany; [2]Werner-Reichardt Centre for Integrative Neuroscience, Tübingen, Germany; [3]Graduate Training Centre of Neuroscience – International Max-Planck Research School (IMPRS), Tübingen, Germany; [4]Chinese Academy of Sciences, Key Laboratory of Brain Connectome and Manipulation, The Brain Cognition and Brain Disease Institute, Shenzhen Institute of Advanced Technology, Chinese Academy of Sciences, Nanshan, China; [5]Shenzhen-Hong Kong Institute of Brain Science-Shenzhen Fundamental Research Institutions, Shenzhen, China

*For correspondence:
patricia.preston@cin.uni-
tuebingen.de (PP-F);
andrea.burgalossi@cin.uni-
tuebingen.de (AB)

Competing interest: The authors declare that no competing interests exist.

**Abstract** Neural circuits are made of a vast diversity of neuronal cell types. While immense progress has been made in classifying neurons based on morphological, molecular, and functional properties, understanding how this heterogeneity contributes to brain function during natural behavior has remained largely unresolved. In the present study, we combined the juxtacellular recording and labeling technique with optogenetics in freely moving mice. This allowed us to selectively target molecularly defined cell classes for in vivo single-cell recordings and morphological analysis. We validated this strategy in the CA1 region of the mouse hippocampus by restricting Channelrhodopsin expression to Calbindin-positive neurons. Directly versus indirectly light-activated neurons could be readily distinguished based on the latencies of light-evoked spikes, with juxtacellular labeling and post hoc histological analysis providing 'ground-truth' validation. Using these opto-juxtacellular procedures in freely moving mice, we found that Calbindin-positive CA1 pyramidal cells were weakly spatially modulated and conveyed less spatial information than Calbindin-negative neurons – pointing to pyramidal cell identity as a key determinant for neuronal recruitment into the hippocampal spatial map. Thus, our method complements current in vivo techniques by enabling optogenetic-assisted structure–function analysis of single neurons recorded during natural, unrestrained behavior.

## Editor's evaluation

This study presents a major technical advance by recording from genetically identified neurons in freely moving mice. This method is applied to the hippocampus to determine circuit specific synaptic interaction in vivo and to compare behavioral correlates of genetically-defined cell types. This technique paves the way for future studies aiming to relate genetics, circuits, and neuronal coding in freely moving animals.

## Introduction

Since the seminal work of Ramón y Cajal it has become clear that neuronal circuits consist of a large variety of neuronal elements. Such diversity encompasses virtually all aspects of neuronal biology, including morphological, genetic, physiological, and functional properties. Over recent years, major

advances have been made toward the systematic classification of neuronal cell types (*Ascoli et al., 2008*; *Scala et al., 2020*; *Tasic et al., 2016*; *Zeisel et al., 2015*). From these research lines, it has become increasingly clear that a combinatorial approach – integrating morphological, molecular, electrophysiological, and functional properties – is required for neuronal cell-type classification.

As inventories of cell types within brain circuits are beginning to emerge, the next challenge is to understand how these cell types contribute to neural circuit computations during behavior. Bridging this gap is a methodological challenge, since current in vivo techniques for neural circuit analysis – for example optogenetic tagging and Ca$^{2+}$ imaging (*Anikeeva et al., 2011*; *Ghosh et al., 2011*; *Lima et al., 2009*) – provide only limited information about the recorded neuronal elements. These methods largely rely upon the genetically restricted expression of Channelrhodopsin (ChR2) or Ca$^{2+}$ indicators – for example via Cre driver lines (*Adesnik, 2018*; *Gafford and Ressler, 2016*; *Kravitz et al., 2013*). In the case of 'optogenetic tagging', recordings are performed with a dual optical and electrical probe (referred to as 'optrode') and short-latency spikes elicited upon illumination provide information about the identity of the recorded units. One limitation of this approach is that 'cell types' are defined by single features, like genetic marker expression or projection target. Additional structural features – like morphology, local and long-range axonal projections, and gene expression profiles – are not accessible with these techniques, thereby making it difficult to map neuronal activity (assessed in vivo) to the multidimensional cell-type classification schemes (being developed in vitro).

In vivo single-cell identification techniques can in principle provide the necessary anatomical resolution for multidimensional classification of the recorded neurons. For example, the juxtacellular method (*Pinault, 1996*; *Pinault, 1994*) allows recording of single neurons in awake behaving animals, along with post hoc morphological analysis and molecular phenotyping of the recorded cells (*Averkin et al., 2016*; *Diamantaki et al., 2018*; *Katona et al., 2014*; *Lapray et al., 2012*; *Tang et al., 2014b*; *Valero et al., 2015*). However, a significant limitation of these methods is that neurons are typically sampled by 'blind' procedures; hence, recordings cannot be targeted to a predefined cell class, which makes it extremely challenging – if not impossible – to efficiently sample sparse neuronal elements within a given circuit.

To circumvent this limitation, an elegant approach has been recently developed by combining single-cell identification techniques with optogenetic tagging (*Katz et al., 2013*; *Muñoz et al., 2014*). This approach enabled ChR2-assisted targeting of genetically defined cell classes; in addition, by means of juxtacellular labeling and post hoc morphological/molecular analysis, it also provided access to the multidimensional neuronal features required for precise cell-type classification (*Muñoz et al., 2017*). In its current form however, the applicability of this method is limited to mechanically stable preparations (e.g., anesthetized or head-restrained animals), thereby preventing structure–function analysis in freely moving animals, engaged in ethologically relevant behaviors.

In the present study, we aimed at bridging this gap by combining the single-cell opto-tagging method (*Katz et al., 2013*; *Muñoz et al., 2014*) with the juxtacellular procedures that we have recently established in freely moving mice (*Diamantaki et al., 2018*). This approach enabled us to efficiently target juxtacellular recording and labeling to predefined, genetically tagged cell classes in awake animals, engaged in natural exploratory behavior.

We demonstrate the technical feasibility of this approach in the dorsal CA1 of the mouse hippocampus. During behavior, the activity of 'place cells' in this region (*O'Keefe, 1976*) is thought to contribute to the encoding of spatial and episodic experiences, and thus to form the neural basis of 'memory engrams' (*Tonegawa et al., 2018*). Anatomically – despite its relatively simple organization – the CA1 has a remarkably diverse cellular composition. In vitro and in vivo work has distinguished more than 20 different morphofunctional interneuronal classes (*Bezaire and Soltesz, 2013*; *Freund and Buzsáki, 1998*; *Klausberger and Somogyi, 2008*), and recent transcriptomic studies point to a seemingly complex heterogeneity among pyramidal neurons, with at least 15 classes being identified purely based on gene expression data (*Yao et al., 2021*). At present it is unclear how these distinct cell types contribute to neural activity during behavior. As test ground for our technique, we focused on a well-defined dimension of pyramidal cell diversity, namely along the radial (deep-superficial) axis of the dorsal CA1 (deep, closer to stratum oriens; superficial, closer to stratum radiatum). Indeed, a growing body of evidence indicates that the anatomical location of the neurons along this axis correlates with in vivo activity patterns (*Cohen et al., 2017*; *Danielson et al., 2016*; *Fattahi et al., 2018*; *Geiller et al., 2017*; *Li et al., 2017*; *Mizuseki et al., 2011*; *Sharif et al., 2021*; *Valero et al.,*

*2015*; for review, see *Preston-Ferrer and Burgalossi, 2018*; *Soltesz and Losonczy, 2018*). We thus took advantage of the expression of Calbindin (*Calb1*) – which is selective for superficially located pyramidal neurons – for testing our juxtacellular opto-tagging procedures in freely behaving mice. We found that, while animals explored a familiar environment, *Calb1*-negative pyramidal neurons were preferentially recruited into the place cell map, while *Calb1*-positive pyramidal cells were only weakly spatially modulated. These data are thus consistent with the emerging idea of an asymmetric recruitment of CA1 pyramidal cells types into the hippocampal representation.

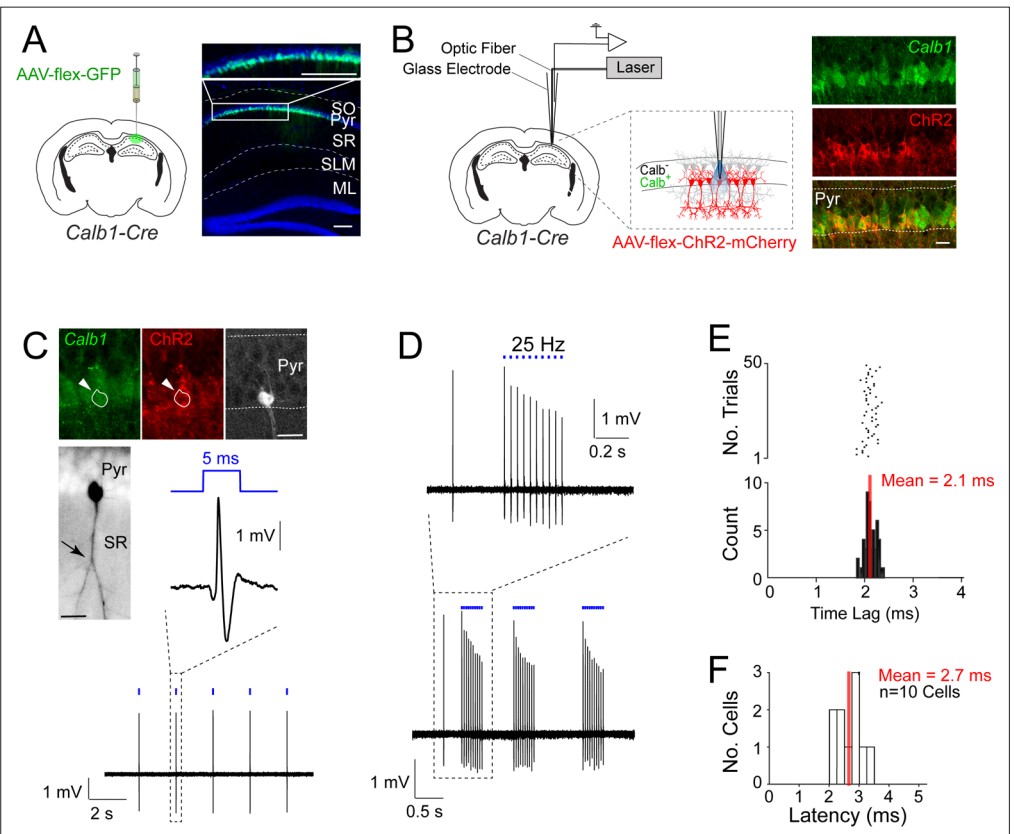

**Figure 1.** Juxtacellular opto-tagging of *Calb1*-positive CA1 pyramidal cells in anesthetized mice. (**A**) Left, a schematic of a mouse coronal brain section showing the injection site of AAV-CAG-flex-GFP in *Calb1*cre mice. Right, epifluoresence image showing Green Fluorescent Protein (GFP) expression (green) in the superficial pyramidal cell layer (blue, DAPI). SO, stratum oriens; Pyr, stratum pyramidale; SR, stratum radiatum; SLM, stratum lacunosum moleculare; ML, molecular layer of dentate gyrus. Scale bar = 100 µm. (**B**) Left, a schematic of a mouse coronal brain section showing the juxtacellular opto-tagging recording configuration. Right, a confocal image showing the expression of mCherry coexpressed with ChR2 (red) in the superficial *Calb1*-positive pyramidal layer (green). Scale bar = 20 µm. (**C**) A representative light-responsive pyramidal neuron, recorded in vivo. Top, epifluorescence images showing *Calb1* (green), ChR2 (red), and Neurobiotin labeling of the neuron (white). Middle, z-stack projection of the same neuron (inverted signal). The arrowhead indicates a branching point of the primary apical dendrite. Bottom, high-pass filtered juxtacellular spike trace, showing short-latency spike responses upon pulses of blue light (5 ms, indicated in blue). Scale bar = 20 µm. (**D**) Bottom, high-pass filtered juxtacellular spike traces of a representative neuron responding to 25 Hz light-pulse stimuli. Top, high-magnification view on a single train (25 Hz, 10 pulses). (**E**) Raster plot (top) and peristimulus time histogram (bottom) showing the spike latency to the light stimulus for the neuron shown in C. The average latency is indicated. (**F**) Histogram of average spike latencies of all identified *Calb1*-positive pyramidal neurons in CA1 (*n* = 10).

The online version of this article includes the following source data and figure supplement(s) for figure 1:

**Source data 1.** Juxtacellular opto-tagging data, average spike latencies (source data for panels E and F).

**Figure supplement 1.** Juxtacellular opto-tagging of *Calb1*-positive interneurons in the CA1 region.

**Figure supplement 1—source data 1.** Electrophysiological properties of putative Calb1-positive interneurons (source data for panels C,E,F,H,I).

# Results

To genetically target *Calb1*-positive CA1 pyramidal neurons, we took advantage of the *Calb1*[cre] driver line (*Daigle et al., 2018*; *Nigro et al., 2018*). As expected from the distribution of *Calb1*-positive neurons within the CA1 pyramidal layer, injections of AAV-CAG-flex-GFP in *Calb1*[cre] mice resulted in selective labeling of superficial pyramidal cells (*Figure 1A*) along with scattered, putative interneurons located primarily within the stratum oriens (not shown).

To record the activity from single *Calb1*-positive CA1 pyramidal neurons, we selectively expressed ChR2 in *Calb1*-positive CA1 neurons by injecting AAV-hSyn1-flex-ChR2-mCherry in the dorsal CA1 of *Calb1*[cre] mice. We then employed the juxtacellular opto-tagging configuration (also known as 'opto-patcher'; *Katz et al., 2013*; *Muñoz et al., 2014*) – consisting of a 50-μm-core diameter optic fiber, connected to a blue laser source, placed inside the glass electrode (*Figure 1B*) – for recording single CA1 neurons in anesthetized animals. Once a recording was established from a putative CA1 pyramidal neuron, low-power (0.1–1 mW) brief blue light pulses (5 ms) were delivered via the optic fiber to test for light-evoked spike responses. Subsequently, the neuron was labeled via juxtacellular procedures. One representative recording is shown in *Figure 1C–E*. This neuron was located in the superficial pyramidal layer and its apical dendrite displayed a proximal branching point (see *Figure 1C*) – a characteristic morphological correlate of superficial CA1 pyramidal neurons (*Bannister and Larkman, 1995*; *Li et al., 2017*). This neuron was confirmed to be positive for the marker *Calb1* as well as ChR2 (*Figure 1C*). Indeed, blue light pulses could reliably drive short-latency spikes (mean latency = 2.1 ± 0.12 ms; *Figure 1D, E*) thus confirming the *Calb1*-positive identity of the recorded cell.

Altogether, we recorded and labeled 19 neurons in anesthetized mice. In all cases where short-latency spiking responses were observed (mean latency <4 ms; *Figure 1F*) the neurons were confirmed to be *Calb1*-positive by immunohistochemical analysis (*n* = 10), therefore validating our opto-tagging strategy. Seven additional nonresponsive neurons were also juxtacellularly labeled and confirmed to be *Calb1*-negative pyramidal cells (not shown). In two cases, we recorded the activity of light-responsive, narrow-waveform neurons (spike peak-to-trough <0.4 ms, see *Ding et al., 2020*; *Preston-Ferrer et al., 2016*) which were positive for *Calb1* immunoreactivity, and were thus classified as putative *Calb1*-positive interneurons (see *Figure 1—figure supplement 1A–C*; these neurons were not included in further analysis).

One known challenge of optogenetic tagging experiments is to reliably distinguish directly versus indirectly excited neurons (e.g., *Beyeler et al., 2016*; *Zutshi et al., 2018*). The short latency of our light-responsive neurons (*Figure 1F*) together with post hoc validation by juxtacellular labeling (*Figure 1C*), provide strong support for direct light activation. To further support these observations and benchmark our opto-tagging strategy, we tested our approach with presynaptic input activation. To this end, we injected AAV-CAG-ChR2-mCherry in the CA3 of wild-type mice (*Figure 2A*). By these means, we obtained strong labeling of CA3 axon terminals (Schaffer collaterals), extending into the CA1 strata radiatum and oriens (*Figure 2B*). We then juxtacellularly recorded from single postsynaptic CA1 pyramidal neurons while simultaneously activating presynaptic CA3 terminals. A representative recording from a light-responsive identified CA1 pyramidal neuron is shown in *Figure 2B–D*. In this neuron, photostimulation at ~1.3 mW was sufficient for inducing suprathreshold spiking (*Figure 2C*), although with lower efficiency and longer latencies (7.4 ± 0.8 ms) as compared to the directly light-activated neurons (see *Figure 1C–F*). At the population level, the response latencies of indirectly light-activated CA1 neurons were significantly longer than those of directly light-activated CA1 *Calb1*-positive cells (direct light activation, mean latency = 2.7 ± 0.3 ms, *n* = 10; indirect light activation, mean latency = 7.0 ± 1.8 ms, *n* = 11; p = 0.0010; *Figure 2E*), thereby further validating our latency-based criteria for cell-type classification.

The above data indicate that synaptically activated neurons can be reliably discriminated from directly light-activated cells, at least when 'weak and unreliable' synapses – like typical cortical synapses, including Schaffer terminals (*Csicsvari et al., 2000*; *Sayer et al., 1990*; *Sayer et al., 1989*) – are activated. As additional challenge, we next asked whether our latency-based criteria would still be valid under conditions of 'strong and reliable' synaptic activation. To this end, we performed a subset of experiments on the hippocampal mossy fiber synapse, which is among the strongest and most reliable connections in the nervous system (*Henze et al., 2002*; *Neubrandt et al., 2018*; *Vyleta et al., 2016*). We selectively expressed ChR2 in dentate gyrus granule cells by injecting AAV-hSyn1-flex-ChR2-mCherry (or AAV-hSyn1-flex-oChIEF-TdTomato) in the dentate gyrus of *Calb1*[cre]

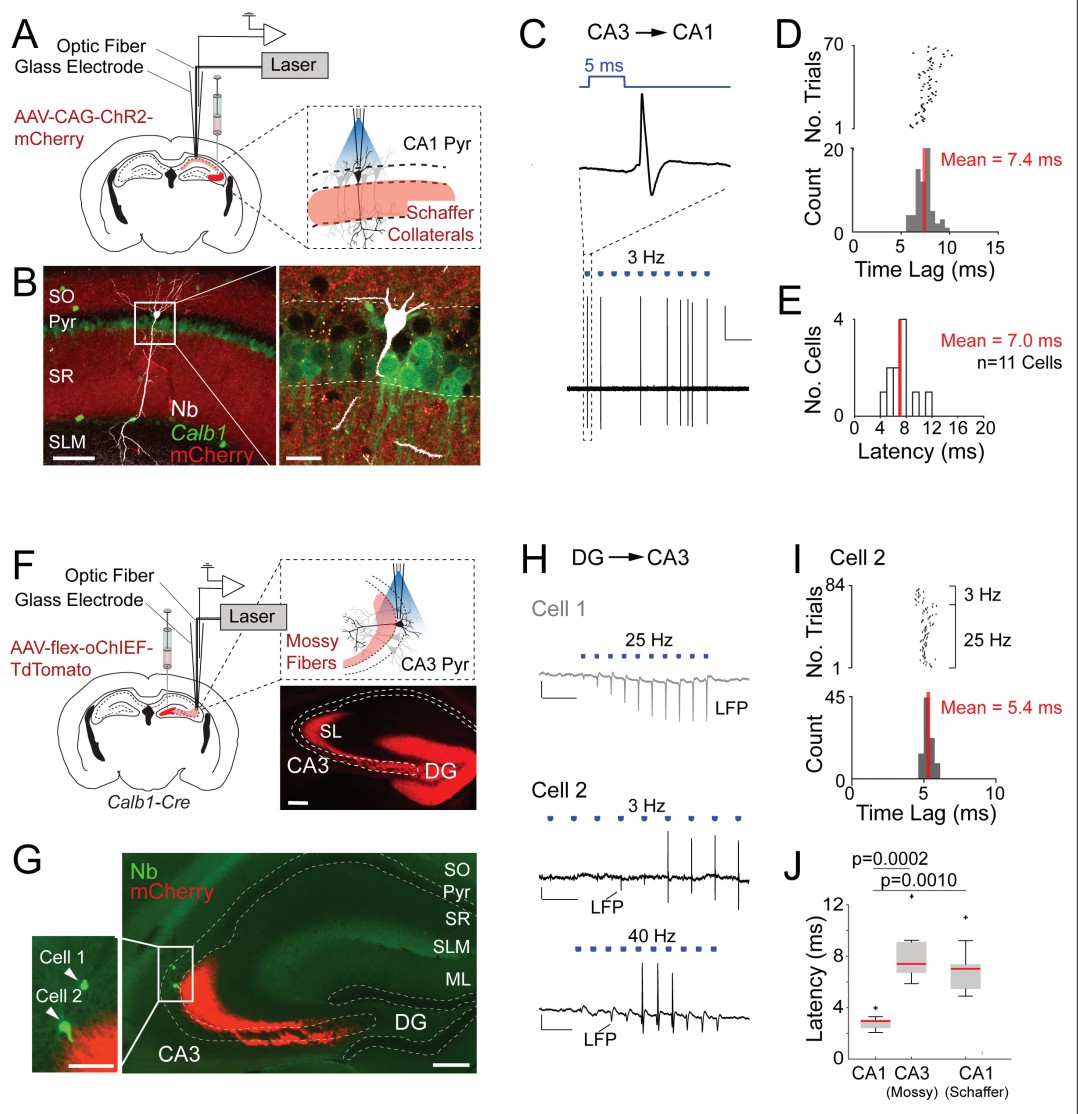

**Figure 2.** Juxtacellular opto-activation of Schaffer collaterals and mossy fiber inputs onto postsynaptic neurons in anesthetized mice. (**A**) Left, schematic of a mouse coronal brain section showing the injection site of an *opsin*-expressing viral vector (AAV-CAG-ChR2-mCherry) in the CA3 region. Right, schematic showing the opto-activation of Schaffer collateral inputs combined with juxtacellular recording from single CA1 pyramidal neurons. (**B**) Confocal images showing a light-responsive CA1 pyramidal neuron (recording shown in C, D) labeled in vivo. Right panel, high-magnification view on the neuron (white, Neurobiotin, Nb) relative to the labeled Schaffer collaterals (red, ChR2-mCherry) and *Calb1* staining (green). Scale bars = 100 μm (left), 20 μm (right). (**C**) High-pass filtered juxtacellular trace (bottom) for the neuron shown in B. Note the partial spiking responses to the light pulses (3 Hz, ~1.3 mW). Top, high magnification showing long-latency spike responses upon pulses of blue light (5 ms, indicated in blue). Scale bar = 2 mV, 2 s. (**D**) Raster plot (top) and peristimulus time histogram (bottom) showing the spike latency to the light stimuli for the CA1 pyramidal neuron shown in B. The average latency is indicated. (**E**) Histogram of average spike latencies of CA1 neurons (*n* = 11) showing spiking responses to Schaffer collateral input activation. The average latency is indicated. (**F**) Left, schematic of a mouse coronal brain section showing the injection site of recombinase-dependent *opsin*-expressing viral vectors (e.g., AAV-EF1a-flex-ChR2-eYFP, AAV-hSyn1-flex-ChR2-mCherry, or AAV-hSyn1-flex-oChIEF-TdTomato) in the dentate gyrus of *Calb1*[cre] mice. Right top, schematic showing the opto-activation of mossy fiber inputs combined with juxtacellular recording from single CA3 pyramidal neurons. Right bottom, epifluorescence image showing eYFP signal (pseudocolored to red for display purposes) following the injection of AAV-CAG-flex-eYFP. Note the labeling of mossy fibers (red) along the transverse CA3 axis. DG, dentate gyrus; SL, stratum lucidum; SO, stratum oriens; Pyr, stratum pyramidale; SR, stratum radiatum; SLM, stratum lacunosum moleculare; ML, molecular layer of dentate gyrus. Scale bar = 200 μm.

*Figure 2 continued on next page*

*Figure 2 continued*

(**G**) Epifluorescence image showing a nonresponsive (Cell 1) and a responsive (Cell 2) CA3a pyramidal neuron, recorded (as in F) along the same electrode penetration and labeled in vivo. Left panel, high magnification on the somata of the two neurons (green, Nb) relative to the labeled mossy terminals (red, oChIEF-TdTomato). Scale bars = 100 µm (left inset), 200 µm (right). (**H**) Representative recordings from the nonresponsive (Cell 1) and responsive (Cell 2) CA3a pyramidal neurons shown in G. The recording form Cell 1 (top, gray) shows the absence of spiking, but increasing amplitude of the negative local field potential (LPF) deflection during the stimulus train (25 Hz, 5-ms pulse duration) – consistent with the expected facilitation of neurotransmitter release at mossy terminals. Scale bars = 2 mV, 100 ms. The recording from Cell 2 (middle and bottom, black) shows partial spiking responses to low and high-frequency light pulses (3 and 40 Hz, 5 ms). Note the increase of the local field potential (LFP) amplitude (indicated with an arrow) during the stimulus trains. Scale bars = 2 mV, 500 ms (middle trace); 2 mV, 50 ms (bottom trace). (**I**) Raster plot (top) and peristimulus time histogram (bottom) showing the spike latency to the light stimuli for Cell 2 shown in G,H. 40 Hz light stimulus trains were excluded because of a high probability of spiking failure (see details in Materials and Methods). The average latency is indicated. (**J**) Boxplots showing comparison of latencies between directly activated CA1 cells ($n$ = 10), indirectly activated CA3 (via mossy fiber photostimulation, $n$ = 7), and CA1 cells (via Schaffer collateral photostimulation, $n$ = 11). Significant p values after multiple group comparison are indicated (Kruskal–Wallis, p = 0.00006). Red lines indicate medians. Outliers are shown as crosses.

The online version of this article includes the following source data for figure 2:

**Source data 1.** Juxtacellular opto-tagging data, average spike latencies (source data for panels D,E and I,J).

mice (*Figure 2F*). By these means, we obtained strong labeling of presynaptic mossy fibers terminals, extending into the CA3 field (*Figure 2F, G*). We then juxtacellularly recorded from single postsynaptic CA3 pyramidal neurons while simultaneously activating presynaptic mossy fibers at different light stimulation frequencies (e.g., 3–40 Hz; see details in Materials and Methods). Two representative recordings from a nonresponsive and a responsive identified CA3 pyramidal neuron are shown in *Figure 2G–I*. In the first neuron, no spiking activity was observed upon light illumination; however, a prominent light-induced facilitation of local field potential amplitude was observed – consistent with the known facilitation of neurotransmitter release at mossy terminal (*Capogna, 1998*; *Vyleta et al., 2016*; *Zucca et al., 2017*). In the second neuron, photostimulation was sufficient for inducing suprathreshold spiking (*Figure 2H*), whose probability increased as a function of stimulus number, consistent with presynaptic facilitation of mossy inputs. Notably, the response latencies of synaptically activated CA3 neurons (mean latency = 8.1 ± 2.4 ms, $n$ = 7) were significantly longer than that of directly light-activated CA1 *Calb1*-positive cells (*Figure 2J*).

Altogether, this dataset shows the technical feasibility of combining the opto-juxtacellular approach for single-cell recording, labeling, and presynaptic input manipulations. Moreover, the longer spike latencies in indirectly activated neurons compared to the directly activated ones (*Figure 2J*) further corroborate the validity of our latency-based opto-tagging approach.

Based on these results, we took advantage of the juxtacellular opto-tagging approach for assessing the in vivo activity patterns of *Calb1*-positive and *Calb1*-negative CA1 pyramidal neurons. To this end, we adapted juxtacellular opto-tagging procedures (*Figure 1*) to freely moving mice. Animals were implanted with miniaturized components (see *Figure 3A*; *Diamantaki et al., 2018*) and a 50-µm-core optic fiber was inserted within the glass electrode. The use of a reversible fiber fixation strategy, together with a mechanical micropositioning system (see *Figure 3—figure supplement 1*; details in Materials and Methods) allowed us to perform multiple recording/opto-tagging trials within individual animals.

Using these juxtacellular opto-tagging procedures, we monitored the activity of single CA1 pyramidal neurons while mice explored a familiar maze. All neurons included in the present study displayed low firing rates (<10 Hz; average firing rate, 2.16 ± 2.06 Hz; $n$ = 54) and often fired complex spikes (average burst index, 0.28 ± 0.18, $n$ = 45) – features classically associated with principal cell identity (*O'Keefe and Dostrovsky, 1971*; *Ranck, 1973*). All histologically identified neurons were classified as pyramidal neurons (see Materials and Methods), thus further confirming our electrophysiological classification criteria.

As in the anesthetized dataset (*Figure 1*), neurons recorded in behaving mice were classified as *Calb1*-positive if short-latency spikes (<4 ms) were reliably evoked by the light pulse ($n$ = 15; *Figure 3B*). In a subset of recordings, juxtacellular labeling was performed and the cytochemical identity of the recorded neuron was assessed via *Calb1* immunoreactivity (*Figure 3C–H*). Neurons were classified as

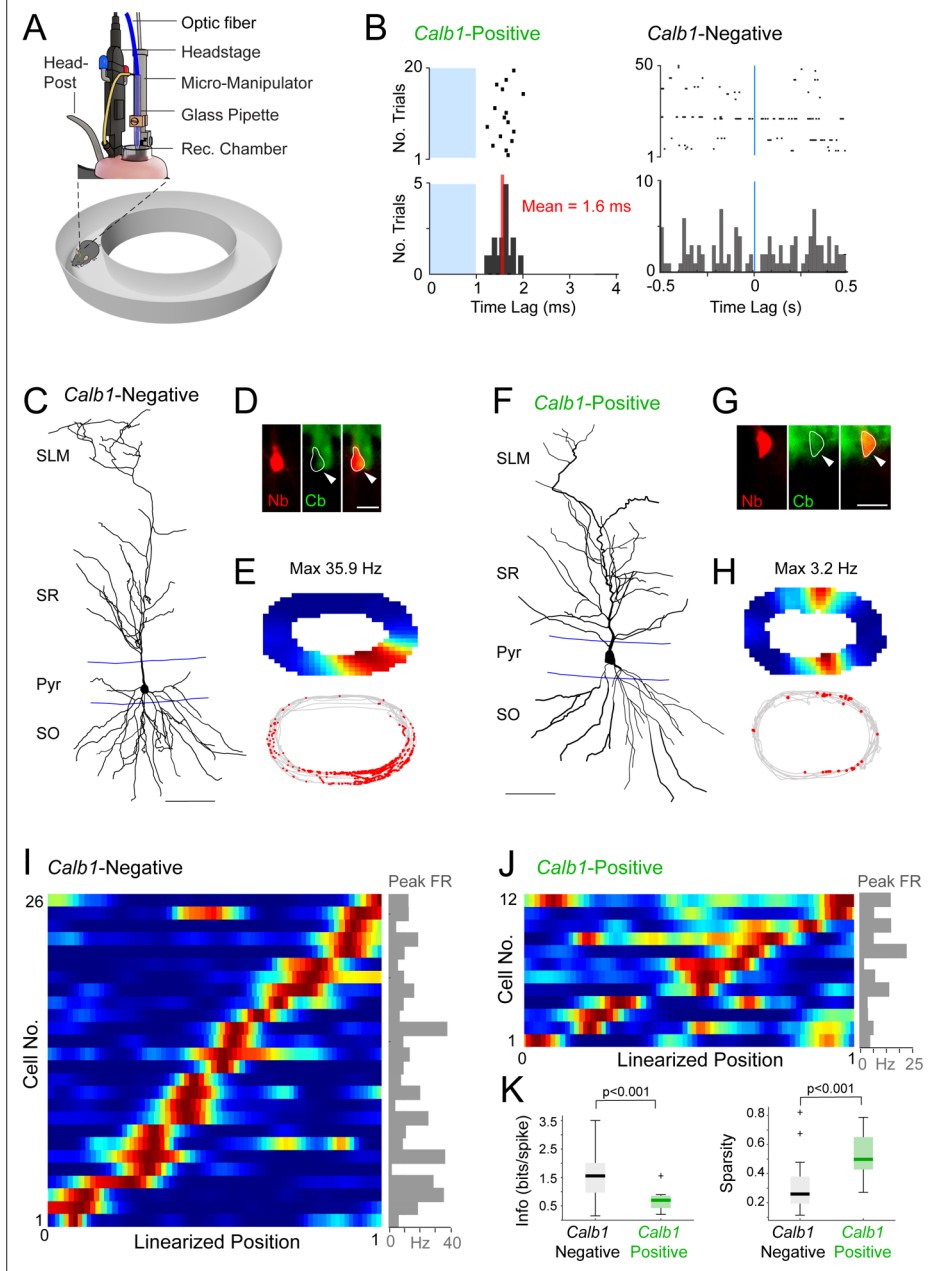

**Figure 3.** Juxtacellular opto-tagging of single CA1 pyramidal neurons in freely moving mice. (**A**) Schematic drawing of the fully assembled recording implant for juxtacellular opto-tagging in freely moving mice (adapted from *Diamantaki et al., 2018*). (**B**) Raster plots (top) and peristimulus time histograms (bottom) showing responses from a *Calb1*-positive (left) and *Calb1*-negative (right) neuron to the light stimulus (indicated in blue). Note the short-latency spiking of the *Calb1*-positive neurons (mean latency indicated) and the absence of response in the *Calb1*-negative cell. (**C**) Reconstruction of the dendritic morphology of a *Calb1*-negative CA1 pyramidal neuron, recorded in a freely moving mouse (recording shown in E). Scale bar = 50 µm. (**D**) Epifluorescence image showing *Calb1* staining (green) and the juxtacellular labeled CA1 neuron (Neurobiotin, red) for the neurons shown in C. Scale bar = 20 µm. (**E**) Rate map (top) and spike-trajectory plot (bottom) for the neuron shown in (**C**). (**F–H**) Same as in (**C–E**) but for a representative *Calb1*-positive CA1 pyramidal neuron. (**I**) Linearized rate maps of all *Calb1*-negative cells (*n* = 26) that met the inclusion criteria for spatial analysis (see Materials and Methods). Each row represents the normalized firing rates relative to the linearized 1D projection of the circular arena. Cells are ordered according to their maximal firing rate position along the linearized trajectory. The bar graph on the right indicates peak firing rates ('Peak FR') for each cell, calculated from the original, non-linearized rate maps. (**J**) Same as in (**I**) except for all *Calb1*-positive cells (*n* = 12) that met the inclusion criteria for spatial analysis (see Materials

*Figure 3 continued on next page*

*Figure 3 continued*

and Methods). (**K**) Boxplots showing spatial information content and sparsity for *Calb1*-negative (*n* = 26, as in I) and *Calb1*-positive (*n* = 12, as in J) CA1 neurons. p values are indicated (Wilcoxon rank-sum test). Black/green lines indicate medians. Outliers are shown as crosses.

The online version of this article includes the following source data and figure supplement(s) for figure 3:

**Source data 1.** Electrophysiological properties of Calb1-positive and Calb1-negative neurons (source data for panels I-K and *Figure 3—figure supplement 2K*).

**Figure supplement 1.** Mechanical micropositioning drive enables multiple juxtacellular recordings within individual animals.

**Figure supplement 1—source data 1.** Average spike waveforms (source data for panels C and E).

**Figure supplement 2.** Stability analysis of spatial firing in *Calb1*-postive and *Calb1*-negative neurons.

---

*Calb1*-negative neurons (*n* = 39) if they did not respond to the light stimulus or were located in the deep pyramidal layer and were negative for *Calb1* expression (see Materials and Methods). In two cases, we recorded from light-responsive, putative *Calb1*-positive interneurons, which fired at high rates during exploration (>10 Hz average firing rates; see *Figure 1—figure supplement 1D–I*; these neurons were not included in further analysis).

During spatial exploration, *Calb1*-positive and *Calb1*-negative pyramidal cells displayed similar average firing rates (*Calb1*-positive, 2.17 ± 2.02 Hz, *n* = 15; *Calb1*-negative, 2.15 ± 2.10 Hz, *n* = 39; p = 0.86) and tendency to fire spike bursts ('burst index'; *Calb1*-positive, 0.29 ± 0.21, *n* = 13; *Calb1*-negative, 0.29 ± 0.18, *n* = 32; p = 0.97). However, these two cell types differed remarkably in the degree of spatial modulation. While the large majority of *Calb1*-negative neurons displayed spatially localized firing patterns, which were homogeneously distributed over the available space (*Figure 3I*), the spiking activity of *Calb1*-positive neurons was less spatially modulated (*Figure 3J*) (*n* = 26 *Calb1*-negative and *n* = 12 *Calb1*-positive neurons included in the spatial analysis; see details in Material and Methods). To quantify these differences, we computed spatial information content (*Skaggs et al., 1993*) and the sparsity index – a metric of how compact the place field is relative to the recording arena (*Jung et al., 1994*). Indeed, we found that *Calb1*-positive neurons conveyed significantly less spatial information (*Calb1*-positive, 0.65 ± 0.35 bits/spike, *n* = 12; *Calb1*-negative, 1.56 ± 0.79 bits/spike, *n* = 26; p = 0.0003) and displayed a higher sparsity index (*Calb1*-positive, 0.53 ± 0.15, *n* = 12; *Calb1*-negative, 0.31 ± 0.17, *n* = 26; p = 0.0003) (*Figure 3K*), indicating a more diffuse firing pattern compared to *Calb1*-negative cells. To further confirm these observations, we performed a spatial stability analysis by calculating the Pearson's correlation coefficient for rate maps computed for the two halves of each recording (*Figure 3—figure supplement 2*). Indeed, in line with their weaker spatial tuning (*Figure 3K*), spatial firing in *Calb1*-positive neurons was significantly less stable than in *Calb1*-negative cells (mean correlation coefficient, *Calb1*-positive, 0.33 ± 0.35; *n* = 12; *Calb1*-negative, 0.68 ± 0.31; *n* = 26; p = 0.0019; see *Figure 3—figure supplement 2*). Altogether, these findings are in line with previous observations (*Danielson et al., 2016*; *Geiller et al., 2017*; *Mizuseki et al., 2011*; *Oliva et al., 2016*; *Sharif et al., 2021*) and indicate that in the dorsal CA1, place cells are preferentially contributed by deep (*Calb1*-negative) pyramidal neurons, while *Calb1*-positive neurons are only weakly spatially modulated.

## Discussion

A mechanistic dissection of neural circuits requires resolving the contribution of individual neuronal elements to in vivo function, neural computation and behavior. Significant experimental efforts have been undertaken, aimed at resolving neuronal diversity within brain circuits. However, due to methodological limitations, it is still largely unknown whether and how the individual neuronal cell classes – typically identified in vitro – contribute to in vivo activity during natural behavior.

The advent of 'genetic tagging' has revolutionized neural circuit research. By restricting the expression of ChR2 (or Ca²⁺ indicators) to a specific subset of neurons, it has become possible to link in vivo activity patterns to the underlying neuronal elements. In freely behaving animals, the combination of extracellular recordings with optogenetic tagging (e.g., *Anikeeva et al., 2011*; *Lima et al., 2009*) has become a gold-standard approach for neural circuit dissection. However, while the implementation of

the 'opto-tagging' method is rather straightforward, there are two important limitations to be considered. First, cell-type identification is indirect, since it is inferred from the spiking responses to the light stimuli. While a number of criteria have been proposed for distinguishing directly versus indirectly light-activated neurons, this distinction is often far from trivial, especially within networks with high recurrent connectivity (*Beyeler et al., 2016*; *Roux et al., 2014*; *Zutshi et al., 2018*). The reliability of optogenetic identification is further complicated by the challenge of 'spike sorting' during periods of highly synchronous activity – occurring for example during photostimulation – since the superimposition of multiple, co-occurring spike waveforms can make the detection of individual spikes difficult and unreliable (*Roux et al., 2014*; *Stark et al., 2012*). While a number of strategies have been proposed for alleviating these problems (*Roux et al., 2014*; *Royer et al., 2010*; *Stark et al., 2012*; *Wu et al., 2015*), obtaining 'ground truth' validation of opto-tagging procedures is particularly challenging, especially in freely behaving animals. A second important limitation is that opto-tagging approaches do not provide access to structural neuronal features – like for example morphology and molecular expression patterns – which are necessary for unequivocal cell-type identification.

In the present study, we aimed at filling these methodological gaps by combining optogenetic tagging with juxtacellular procedures in freely moving mice. This combination of techniques enables targeting of juxtacellular recordings to a genetically predefined cell class; hence, it represents a significant advance over current juxtacellular protocols, where individual neurons are blindly sampled within a target structure (*Averkin et al., 2016*; *Diamantaki et al., 2018*; *Katona et al., 2014*; *Lapray et al., 2012*; *Tang et al., 2014a*; *Valero et al., 2015*). We provide proof-of-principle validation of our method by recording from *Calb1*-positive pyramidal neurons (*Figure 3*) in the mouse CA1. In a limited number of cases, we monitored the activity of *Calb1*-positive interneurons, whose somata were located outside of the pyramidal layer (*Figure 1—figure supplement 1*). These neurons fired at high rates during spatial exploration, and did not display the 'fast-spiking' phenotype which is often associated to perisomatic-targeting CA1 interneurons ('theta cells'; *O'Keefe and Dostrovsky, 1971*; *Ranck, 1973*). This evidence indicates that our juxtacellular opto-tagging approach could facilitate the targeting of sparse neuronal populations – like different subtypes of GABAergic neurons (*Bartos et al., 2011*; *Bartos and Elgueta, 2012*; *Klausberger, 2009*; *Klausberger and Somogyi, 2008*) – whose blind sampling in freely moving animals, along with post hoc immunohistochemical phenotyping, is known to be very challenging and labor intensive (*Averkin et al., 2016*; *Katona et al., 2020*; *Lagler et al., 2016*; *Lapray et al., 2012*).

Our combined opto-juxtacellular method also provides several advantages over 'conventional' extracellular opto-tagging approaches. First, it enables direct validation of opto-tagging criteria, in that single ChR2-responsive (or nonresponsive) neurons can not only be recorded, but also labeled by juxtacellular procedures, thereby enabling post hoc verification of ChR2 and/or marker expression. Second, the large amplitude juxtacellular spikes (typically >1–2 mV peak-to-peak amplitudes, *Herfst et al., 2012*; *Tang et al., 2014a*) allow unequivocal spike identification, thereby providing 'ground truth' spiking signals that are not dependent on the performance of spike sorting algorithms, especially during photostimulation periods. Third, post hoc morphological and molecular expression analysis of the recorded/labeled cells provides access to structural features that are typically not accessible with alternative techniques, thereby enabling unequivocal cell-type classification. Moreover, our data also provide proof-of-principle evidence that our juxtacellular recording/labeling procedures can also be used in combination with presynaptic input manipulations (*Figure 2*). This approach can be particularly useful for mapping synaptic inputs onto identified cell types, like for example the distinct subtypes of CA2 and CA3 pyramidal neurons (*Helton et al., 2019*; *Hunt et al., 2018*; *Marissal et al., 2012*; *Raus Balind et al., 2019*) or dentate granule cells (*Diamantaki et al., 2016*; *Erwin et al., 2020*; *Zhang et al., 2020*) which have been classified based on dendritic morphological criteria. We envision that the future combination with single-cell stimulation (*Diamantaki et al., 2018*; *Diamantaki et al., 2016*), will allow pairing between pre- and postsynaptic activity (*Nicoll and Schmitz, 2005*; *Salin et al., 1996*; *Toth et al., 2000*), thus enabling the study of single-cell plasticity mechanisms, so far limited to simplified preparations, to the intact system during natural behaviors.

While it is technically possible to perform multiple opto-juxtacellular recordings within individual experiments (*Figure 3—figure supplement 1*), our method still remains laborious and of limited output compared to alternative techniques (e.g., extracellular recordings). Hence, it should not be seen as high throughput, but rather as complementary to existing in vivo recording methods. We

envision that opto-juxtacellular datasets – although limited – could still provide a valuable complement to current cell-type classification approaches, for example by offering the possibility of building a classifier, which could then be used for assigning cell identity to 'blind' extracellular units (as in e.g. *GoodSmith et al., 2017*; *Tang et al., 2014b*).

Recent in vitro and in vivo work has shown that the CA1 pyramidal layer can be subdivided into two sublayers (superficial and deep) according to morphological, molecular, and electrophysiological criteria (*Baimbridge and Miller, 1982*; *Slomianka et al., 2011*; *Preston-Ferrer and Burgalossi, 2018*; *Soltesz and Losonczy, 2018*). Deep and superficial pyramidal cells have been shown to display distinct in vivo activity patterns relative to local field potential oscillations (*Navas-Olive et al., 2020*; *Valero et al., 2015*) and distinct tendencies to express place fields (*Danielson et al., 2016*; *Geiller et al., 2017*; *Mizuseki et al., 2011*; *Navas-Olive et al., 2020*; *Sharif et al., 2021*). However, most of the previous in vivo observations were based upon the anatomical assignment of extracellular recording locations (but see *Valero et al., 2015*; *Navas-Olive et al., 2020*). As a testing ground for our technique, we recorded, opto-tagged, and labeled single CA1 pyramidal neurons in freely moving mice – thus validating and extending previous observations by mapping functional deep-superficial gradients onto the underlying neuronal elements. We found that *Calb1*-positive pyramidal neurons were only weakly spatially modulated, while 'place cells' were preferentially recruited from the *Calb1*-negative population (*Figure 3*). While this finding is in line with previous work (e.g., *Danielson et al., 2016*; *Geiller et al., 2017*; *Mizuseki et al., 2011*; *Navas-Olive et al., 2020*; *Sharif et al., 2021*), we acknowledge that it rests on a limited number of observations (*Figure 3I–K*). Notably, inputs from the medial and lateral entorhinal cortices are known to preferentially target deep and superficial CA1 pyramidal cells, respectively (*Li et al., 2017*; *Masurkar et al., 2017*); hence, the weaker spatial selectivity of upstream lateral entorhinal inputs (*Hargreaves et al., 2005*) might account for the weaker spatial tuning of *Calb1*-positive CA1 pyramidal neurons.

In summary, we demonstrated the technical feasibility of performing ChR2-assisted juxtacellular recordings in freely moving mice. We envision that this method will be useful for resolving neuronal heterogeneity beyond 'single-marker' classification schemes, thus providing a valuable complement to current techniques for structure–function analysis of neuronal circuits during natural behaviors.

## Materials and methods
### Experimental subjects
Wild-type C57BL/6J mice (RRID:IMSR_JAX:000664; male, >8 weeks old; Charles River, Sulzfeld, Germany) and *Calb1*cre mice (RRID:IMSR_JAX:028532; male, >8 weeks old; The Jackson Laboratory, Bar Harbor, United States, Cat#028532) were used in this study (see details below).

### Viral injections and juxtacellular recordings
AAV-CAG-flex-GFP (*Chan et al., 2017*; Cat#BA-002, Charitè Viral Vector Core, Berlin, Germany; RRID:Addgene_28304) or AAV-hSyn1-flex-ChR2(H134R)-mCherry (Cat#v332-1, Viral Vector Facility, University of Zürich, Zurich, Switzerland; RRID:Addgene_20297) were injected in the dorsal CA1 of *Calb1*cre mice (2.0 mm posterior to bregma; 1.8 mm lateral to midline). AAV-CAG-ChR2-mCherry (Addgene, Cat#100054; RRID:Addgene_100054) was injected in the dorsal CA3 of C57BL/6J mice (2.0 mm posterior to bregma; 2.0 mm lateral to midline). AAV-EF1a-flex-ChR2-eYFP (Addgene, Cat#20298-AAV9, Watertown, MA; RRID:Addgene_20298), AAV-hSyn1-flex-ChR2(H134R)-mCherry, AAV-hSyn1-flex-oChIEF-TdTomato (Charitè Viral Vector Core, Cat#BA-030; RRID:Addgene_30541), and AAV-CAG-ChR2-mCherry were injected in the dorsal dentate gyrus of *Calb1*cre mice or C57BL/6J mice (2.2 mm posterior to bregma; 1 mm lateral to midline). Viral injections were performed via pressure injection, essentially as previously described (*Kitamura et al., 2014*; *Liu et al., 2012*). Briefly, animals were anesthetized with fentanyl-midazolam-medetomidine anesthesia (*Burgalossi et al., 2011*; *Chakrabarti and Schwarz, 2018*). 50–150 nl of viral solution (titer ~$10^{11}$–$10^{12}$ vg/ml) were injected with a pression-injection pump (Microinjection Syringe Pump, Cat#UMP3T, WPI) mounted onto a Robot Stereotaxic (StereoDrive, Cat#9001001 Neurostar). Twelve to eighteen days following injection, mice were subjected to histological analysis (see paragraph 'Histology, immunohistochemistry and neuronal reconstructions') or electrophysiological recordings (see paragraph 'Opto-tagging procedures'). In *Calb1*cre mice injected within the CA1 pyramidal layer, ChR2 was almost exclusively

expressed in *Calb1*-positive neurons (0 out of 272 neurons were ChR2-positive/*Calb1*-negative), thus confirming the specificity of the *Calb1*^cre driver line (see also *Daigle et al., 2018*; *Nigro et al., 2018*). Viral titers and injection volumes were optimized to obtain high infection efficiency and ChR2 expression levels (~95.8% of *Calb1*-positive neurons were also ChR2-positive; 261/272 neurons), thereby minimizing the occurrence of false negatives in our opto-tagging experiments.

Experimental procedures for obtaining juxtacellular recordings, signal acquisition and processing were essentially performed as described previously (*Diamantaki et al., 2018*; *Tang et al., 2014a*). Briefly, glass electrodes with resistance 4–6 MΩ were filled with 1.5–2% Neurobiotin (Vector Laboratories; Cat# SP-1120, RRID:AB_2313575) in Ringer's solution containing (in mM): 135 NaCl, 5.4 KCl, 5 HEPES, 1.8 CaCl$_2$, and 1 MgCl$_2$ or Intracellular solution containing (in mM): 135 K-gluconate, 10 HEPES, 10 Na$_2$-phosphocreatine, 4 KCl, 4 MgATP, and 0.3 Na$_3$GTP. Osmolarity was adjusted to 280–310 mOsm. Before electrophysiological recording (either in anesthetized or awake animals), mapping experiments were performed to precisely estimate the location of dorsal CA1 or CA3. The occurrence of sharp-wave ripples complexes, their polarity reversal and the increased juxtacellular hit rates served as reliable electrophysiological signatures for precisely localizing the CA1 or CA3 pyramidal layer. Histological analysis confirmed the expected electrode location, since all identified cells were located within CA1 or CA3 region respectively (*n* = 38).

For recordings in anesthetized animals, mice were anesthetized with ketamine/xylazine, as previously described (*Diamantaki et al., 2018*). Recordings were performed with the optopatcher (A-M systems, Maulbronn, Germany, Cat#667844, *Katz et al., 2013*; *Muñoz et al., 2014*) and a 50-µm-core diameter fiber (Thorlabs, Newton, New Jersey, Cat#FG050LGA). Recordings in freely moving animals were obtained by means of a miniaturized micromanipulator (Nanomotor, RRID:SCR_016100; Kleindiek Nanotechnik) secured onto a custom-made base, and an ELC miniature headstage (RRID:SCR_016102) (connected to the preimplanted pin connector), essentially as previously described (*Diamantaki et al., 2018*). In a subset of recordings, we employed a novel custom-made micromanipulator base with a mechanical micropositioning system that allowed adjustments of the electrode position (entry point) between individual sessions (*Figure 3—figure supplement 1*). The location of the animal was tracked using two LEDs (red and blue) mounted on the mouse's head. Prior recordings, an optic fiber (Thorlabs, Cat#FG050LGA) was inserted in the glass electrode and secured with a reversible fixation strategy, that is by means of silicone sealant (Kwik-Cast silicone sealant, WPI). The use of a reversible optic fiber fixation strategy, together with a mechanical micropositioning system of the glass electrode, allowed us to perform multiple recording/opto-tagging trials within individual animals. Recordings were performed while animals were freely exploring an O-shaped, linear maze (70 × 50 cm, 9-cm wide path).

For both anesthetized and freely moving juxtacellular recordings, the juxtacellular voltage signal was acquired via an ELC headstage (Cat#ELC-03XS, anesthetized animals) or an ELC miniature headstage (Cat#ELC-MINI-DIFF-LED, freely moving animals), and an ELC-03XS amplifier (NPI Electronic, Tamm, Germany), sampled at 20 kHz by a POWER1401-3 analog-to-digital interface under the control of Spike2 Software (CED, Cambridge, UK). Juxtacellular labeling was performed according to standard procedures (*Pinault, 1996*; *Pinault, 1994*) with 200-ms-long squared current pulses.

## Opto-tagging procedures

During juxtacellular recording in anesthetized and freely moving mice, an optic fiber (50 µm core) was inserted in the glass pipette. The optical fiber was connected to a 473 nm laser (Oxxius, Cat#L-BX-473–300) and controlled by the Spike2 software. The laser output (controlled by the Oxxius software) was calibrated and measured at the glass electrode tip (containing Ringer solution). We did not systematically test photostimulation intensities for all recorded neurons, but light intensities were adjusted on a cell-by-cell basis to the amount that elicited reliable spiking. Neurons, especially nonresponsive ones, were occasionally tested with laser power intensities up to 5 mW, like the example Cell 1 shown in *Figure 2H* (light stimulation trials with >5 mW output power were not included in the analysis). For directly light-activated CA1 neurons, low-power intensities were sufficient for eliciting reliable spiking (typically <1 mW; range 0.1–1 mW). In the datasets of synaptically activated CA1 and CA3 neurons (*Figure 2*), higher light power (>1 mW; range 1–2.5 mW) was necessary for evoking spiking from light-responsive neurons – consistent with the notion that higher stimulus intensities are needed for synaptic activation, as compared to direct light activation.

Individual juxtacellularly recorded neurons were tested for light-evoked spiking by delivering short pulses of blue light at the end of the recording session (1- to 5-ms long pulses). Notably, while in anesthetized animals, 5-ms light pulses largely evoked single spikes in the responsive CA1 neurons (mean number of evoked spikes per pulse, 1.02 ± 0.04), during awake behavior, the same stimulation protocol reliably evoked burst activity (i.e., 2–3 spikes at Interspike Intervals (ISI) <6 ms; not shown). In order to elicit minimal light-evoked responses in the awake state, pulse duration was thus reduced to 1 ms (1 Hz frequency, 10–50 total pulses; mean number of evoked spikes per pulse, 1.56 ± 0.87). In a subset of anesthetized recordings ($n$ = 4), putative *Calb1*-positive neurons could also be reliably driven to spike with a 100-μm-core optic fiber, coupled to a 470-nm LED light source (M470F3, ThorLab). Spike latencies tended to be slightly longer than – but not significantly different from – spike latencies assessed with laser stimulation (mean latencies; laser: 2.45 ± 0.43 ms; LED: 2.96 ± 0.30 ms; $p$ = 0.11); hence, these recordings were pooled together.

Peristimulus time histograms (PSTHs) were computed with 0.1 ms bin size (the only exception being the putative *Calb1*-negative cell in *Figure 3B*, where the PSTH was computed with a bin size of 20 ms for display purposes). Spike latencies to light stimulations were computed by measuring, for each trial, the time between the onset of the light stimulus and the first evoked spike within a 20-ms window. The mean latency for each cell was calculated as the average spike-latency across all trials (only pulses with interpulse intervals ≥200 ms where included in the analysis of directly light-activated CA1 neurons). A cell was defined as directly light activated (i.e., putative *Calb1*-positive) if short-latency spikes were evoked by the light stimulus (mean latency <4 ms). ChR2-expressing neurons could be reliably entrained by the light stimuli up to 40 Hz frequencies (e.g., see *Figure 1C, D* and *Figure 1—figure supplement 1B*). The average number of light-evoked spikes was calculated by averaging the number of spikes in a 20-ms-long window following the onset of the light stimulus.

## Histology, immunohistochemistry, and neuronal reconstruction

For histological processing, animals were euthanized with an overdose of pentobarbital and perfused transcardially with 0.1 M phosphate-buffered saline followed by a 4% paraformaldehyde solution. Brains were sliced on vibratome (VT1200S; Leica, Wetzlar, Germany) to obtain: 70-μm-thick sagittal or coronal sections.

Immunostainings were performed on free-floating sections as described previously (*Ray et al., 2014*). To reveal the morphology of juxtacellularly labeled CA1 or CA3 cells (i.e., filled with Neurobiotin), brain slices were processed with streptavidin-546 (Thermo Fisher Scientific, Waltham, MA, Cat#S1225) or streptavidin-488 (Thermo Fisher Scientific, Cat#S1223). Immunohistochemical stainings for Calbindin1 were performed with a rabbit anti-Calbindin antibody (Swant, Cat#300; RRID:AB_10000347). Fluorescence images were acquired by epifluorescence microscopy (Axio imager; Zeiss, Jena, Germany) and confocal microscopy (LSM 900; Zeiss). For cell reconstruction, after fluorescence images were acquired, the Neurobiotin staining was converted into a dark DAB reaction product followed by $Ni^{2+}$-DAB enhancement protocol (*Klausberger et al., 2003*). Neuronal reconstructions were performed manually on DAB-converted specimens with Neurolucida software (MBF Bioscience, Williston, Vermont), and displayed as 2D projections.

## Analysis of electrophysiology data

Spike signals from juxtacellular voltage traces were manually isolated, as described previously (*Burgalossi et al., 2011*). Recordings (or portions of recordings) in which cellular damage was observed (e.g., spike-shape broadening, increase in firing rate accompanied by negative DC shifts of the juxtacellular voltage signal, as described in *Pinault, 1996*; *Herfst et al., 2012*) were excluded from the analysis. In the present study (in line with previous work; *Epsztein et al., 2011*; *Lee et al., 2012*; *Diamantaki et al., 2016*; *Diamantaki et al., 2018*), a linear circular maze was used for enabling the assessment of spatial modulation in CA1 pyramidal neurons within shorter recording durations compared to extracellular recordings. Recording displayed in *Figure 3I, J* were converted to a one-dimensional representation. This was done by first projecting the *X–Y* coordinates onto the ellipse that best approximated the trajectory, and then converting the projected coordinates into a one-dimensional representation by finding their associated positions along the linearized ellipse.

The dataset of directly light-activated CA1 recordings in anesthetized animals (*Figure 1*; six mice) consisted of 10 *Calb1*-positive and 7 *Calb1*-negative pyramidal neurons (two additional recordings

from putative *Calb1*-positive interneurons were excluded from further analysis, *Figure 1—figure supplement 1A–C* ). The datasets of indirect (synaptic) activation consisted of 7 light-activated CA3 cells (4 mice) and 11 light-activated CA1 cells (3 mice) (*Figure 2*). The dataset of CA1 recordings in freely moving animals (54 recordings) consisted of 10 recordings from histologically identified neurons (3 of which were reported in a previous study, *Diamantaki et al., 2018*) and 44 recordings from opto-tagged neurons (10 mice). Out of the 10 histologically identified neurons, 2 neurons were classified as *Calb1*-positive and 8 neurons were classified as *Calb1*-negative (by Calbindin1 immunoreactivity and/or their relative position within the pyramidal layer, i.e., deep versus superficial). Of the 44 opto-tagged recordings, 13 were classified as putative *Calb1*-positive and 31 as putative *Calb1*-negative neurons, based on spiking probability and spike latency to the light stimulus (see *Figure 3B* and main text). Two additional recordings from putative *Calb1*-positive interneurons were excluded from further analysis (*Figure 1—figure supplement 1D–I*). Data from one mouse were not included in the analysis because juxtacellular labeling was not performed, and the recording location in the CA1 region could not be unequivocally verified.

For the analysis of spatial firing patterns (spatial information per spike, sparsity, and spatial stability, *Figure 3K* and *Figure 3—figure supplement 2*) a speed threshold was applied (speed >1 cm/s) and only recordings with >50 spikes and ≥3 laps were included. In total, 38 of 54 recordings met these inclusion criteria (12 putative *Calb1*-positive and 26 putative *Calb1*-negative neurons, see *Figure 3I–K*). Applying more stringent inclusion criteria (e.g., >3 laps, > 100 spikes) and/or different speed thresholds (1–10 cm/s) led to qualitatively similar results and did not change the statistical significance of our findings ($P < 0.05$ for *Figure 3I–K*). The burst index was defined as the sum of spikes with an ISI ≤6 ms, divided by the total number of spikes (*Hunt et al., 2018*; *Neunuebel and Knierim, 2012*). Only recordings with >50 spikes were included in the burstiness analysis.

## Analysis of spatial modulation

The position of the mouse was defined as the midpoint between two head-mounted LEDs. For computing color-coded firing rate maps, only spikes during movement (>1 cm/s) were included. Space was discretized into pixels of 2.5 × 2.5 cm, for which the occupancy ($z$) of a given pixel $x$ was calculated as

$$z\left(x\right) = \sum_t w\left(|x - x_t|\right)\Delta t$$

where $x_t$ is the position of the mouse at time $t$, $\Delta t$ the interframe interval, and $w$ a Gaussian smoothing kernel with $\sigma = 1$. Then, the firing rate ($r$) for a given pixel ($x$) was calculated as

$$r\left(x\right) = \frac{\sum_i w\left(|x - x_i|\right)}{Z(x)}$$

where $x_i$ is the position of the mouse when spike $i$ was fired. The firing rate of pixels, whose occupancy ($z$) was less than 20 ms, was not shown.

The spatial information of a cell in bits per spike (*Markus et al., 1994*) is calculated as

$$I_{spike} = \sum_n \left(p_n * (\tfrac{\lambda_n}{\lambda}) * \log_2(\tfrac{\lambda_n}{\lambda})\right)$$

The sparsity index of a cell (*Markus et al., 1994*) is calculated as

$$Sparsity = \sum_n \frac{p_n * \lambda_n^2}{\lambda^2}$$

In both cases, $p_n$ is the probability of the animal being in $n$th pixel bin, $\lambda_n$ is the mean firing rate in the $n$th pixel bin, and $\lambda$ is the overall mean fire rate of the cell.

The stability of spatial firing was quantified by computing rate-map correlation (via Pearson's correlation coefficient) (as in e.g. *Agnihotri et al., 2004*; *Grieves et al., 2020*; *Preston-Ferrer et al., 2016*) between linearized rate maps from the first and second half of each recording.

## Statistical analysis

Statistical analysis was performed with standard MATLAB functions. No statistical methods were used to predetermine sample sizes, but our sample size estimates were based upon previous work

addressing structure–function relationships of single neurons with similar techniques (e.g., *Diamantaki et al., 2016*; *Preston-Ferrer et al., 2016*; *Tang et al., 2014b*, *Tang et al., 2015*). Data are presented as mean ± SD, unless indicated otherwise. Statistical significance was assessed by a two-sided Wilcoxon rank-sum test with 95% confidence (for the comparison of spatial firing patterns between *Calb1*-positive and *Calb1*-negative populations). For multiple groups comparisons (*Figure 2J*), Kruskal–Wallis test was used.

## Acknowledgements

This work was supported by the Werner Reichardt Centre for Integrative Neuroscience (EXC 307), the Eberhard-Karls University of Tübingen, the DFG grant BU 3126/2-1, and a Shenzhen City Grant (JCYJ20180507182458694). We thank Alexandra Eritja and Fabio Monteiro for their excellent assistance with anatomy experiments, Klaus Vollmer and the UKT workshop for excellent support with fine mechanics.

## Additional information

### Funding

| Funder | Grant reference number | Author |
| --- | --- | --- |
| Deutsche Forschungsgemeinschaft | BU 3126 /2-1 | Andrea Burgalossi |
| Shenzhen city grant | JCYJ20180507182458694 | Robert Naumann |
| University of Tübingen | Athene Grant | Patricia Preston-Ferrer |
| University of Tübingen | Intramural funding | Andrea Burgalossi Patricia Preston-Ferrer |

The funders had no role in study design, data collection, and interpretation, or the decision to submit the work for publication.

### Author contributions

Lingjun Ding, Data curation, Formal analysis, Investigation, Methodology, Writing – review and editing; Giuseppe Balsamo, Investigation, Methodology, Writing – review and editing; Hongbiao Chen, Formal analysis, Writing – review and editing; Eduardo Blanco-Hernandez, Formal analysis, Validation, Visualization; Ioannis S Zouridis, Robert Naumann, Investigation, Writing – review and editing; Patricia Preston-Ferrer, Andrea Burgalossi, Conceptualization, Funding acquisition, Project administration, Supervision, Writing - original draft

### Author ORCIDs

Lingjun Ding http://orcid.org/0000-0002-9183-7062
Hongbiao Chen http://orcid.org/0000-0002-0025-4380
Ioannis S Zouridis http://orcid.org/0000-0002-4032-3723
Andrea Burgalossi http://orcid.org/0000-0003-0039-3599

### Ethics

All experimental procedures were performed according to German guidelines on animal welfare under the supervision of local ethics committees (permit numbers CIN8/14, CIN1/17, CIN5/19G, CIN03/20G) and in accordance with the guidelines of the Animal Care and Use Committees at the Shenzhen Institute of Advanced Technology (SIAT), Chinese Academy of Sciences (CAS), China (permit numbers SIAT-IRB-171016-NS-NAUMANN-A0383; SIAT-IACUC-20200901-NS-NBJZX-ROBERT NAUMANN-A0608-02).

### Decision letter and Author response

Decision letter https://doi.org/10.7554/eLife.71720.sa1
Author response https://doi.org/10.7554/eLife.71720.sa2

## Additional files

### Supplementary files
• Transparent reporting form

### Data availability
Data generated during this study are included in the manuscript and supporting files. Source data files have been provided for all figures, i.e. Figure 1, Figure 2, Figure 3, Figure 1- figure supplement 1, Figure 3- figure supplement 1.

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
