## [Editor Report]

This study presents a major technical advance by recording from genetically identified neurons in freely moving mice. This method is applied to the hippocampus to determine circuit specific synaptic interaction in vivo and to compare behavioral correlates of genetically-defined cell types. This technique paves the way for future studies aiming to relate genetics, circuits, and neuronal coding in freely moving animals.

---

## [Decision Letter]

**Decision letter after peer review:**

Thank you for submitting your article "Juxtacellular opto-tagging of hippocampal CA1 neurons in freely-moving mice" for consideration by *eLife*. Your article has been reviewed by 3 peer reviewers, and the evaluation has been overseen by a Reviewing Editor and Laura Colgin as the Senior Editor. The reviewers have opted to remain anonymous.

Essential revisions:

While all three reviewers were impressed by the technical tour-de-force presented in this manuscript, they also agreed that the last part of the study regarding cFos expression is underpowered and would be better suited in a future study focusing on this particular question.

1) The main comment is that the authors should reframe their study as a technical report only, focusing on the combination of opto-tagging (including cell type and synaptic interaction) and juxtacellular recording in freely moving animals. In other words, the study should only include the first three figures, and the abstract should be rephrased to only present the technical aspect of the work. The study should then provide an in-depth discussion of the potential advantages and challenges of combining the two techniques (this is only briefly discussed in the present manuscript).

2) In the present form of the manuscript, it is not immediately clear how Figure 2 relates to the rest of the study, which won't be the case if this the study is framed as a technical report only. The disambiguation between direct and indirect light-evoked spikes is interesting, but it would have been more natural to carry out such experiments in the CA3/CA1 pathway. Some justification of this choice, if possible, would be helpful to the reader.

3) Firing rate is normalized in figure 3I and 3J – but it would be interesting to know what the max firing rate is for each cell (just add the Hz to each row).

*Reviewer #1:*

In this manuscript by Ding et al. from the group of Andrea Burgalossi, the authors use a sophisticated combination of juxtacellular recordings and optogenetics to study the spatial firing properties of genetically defined neurons in CA1 of the hippocampus. They first focus on showing that calbindin-positive pyramidal cell were weakly spatially tuned compared to calbindin-negative pyramidal neurons. The authors finish with a brief analysis showing that cFos expressing cells were often calbindin-expressing and this increased with spatial exploration. Overall this is an impressive body of work and will be informative to those interested in heterogeneity in CA1 pyramidal cells, engram recruitment, and more generally this paper established an important technique that can be applied throughout the brain.

Overall enthusiasm for the paper is somewhat muted given the finding from the McHugh lab that cfos engram cells are not highly spatial, however the novelty here is the additional link to calbindin expression in addition to the novelty and impressive nature of the methodology used.

1. One concern I have is the low number of neurons in the Calbindin positive group (n=7). Of these 7 cells, at least 3 or 4 of them seem as spatial as the calbindin negative group. I realize these are hard experiments, and I don't think more data is needed – but perhaps a more thorough analysis of the data is needed (lap to lap variability, control for firing rates (see below)), etc.

2. It also appears that the firing rates of calbindin-positive neurons is lower (at least from the example in figure 3F this seems to be the case. I think the authors will need to subsample spiking from the calbindin-negative group to match the low firing rate of the calbindin positive population and then compute information and sparsity (I understand these things so account for firing rate, but a subsampled analysis would help to void this concern)).

*Reviewer #2:*

The manuscript of Ding et al. is primarily designed as a technical study, describing the synthesis of two challenging approaches that have hitherto have not been attempted in a single preparation. The main experiments describe the bringing together of in vivo juxtacellular recording of mouse hippocampal CA1 pyramidal cells and cell-type specific optoID in behaving mice. Together these approaches allow the generation of ground truth data on in vivo physiology across the levels of anatomy, genetic marker expression, and anatomical location. They demonstrate the feasibility of the approach both in anesthetized and freely moving mice, as well as providing additional data in Figure 2 speaking to the differences in light-evoked spike latency between direct stimulation of ChR2 expression neurons and neurons post-synaptic to stimulated ChR2 expressing axon terminals. These is a nice confirmation on the specificity of the techique although these data would have been stronger if generating in the same circuit (CA3 presynaptic expression of ChR2 instead of the DG/CA3 circuit).

The authors validate the approach in the CA1 region of the hippocampus, addressing the previously noted distinction in spatial coding between superficial (calbindin positive) and deep (calbindin negative) CA1 pyramidal cells (see studies by Danielson et al. 2016, Mizuseki et al. 2011, Fernandez Ruiz et al. 2017Valero et al. 2015). The data included reinforce the previous findings they reference, but offer no new insights.

1. Finally, a final experiment uses the cFos-tTA mouse to label neurons expressing cFos ('engram neurons ') following exploration of a novel environment to examine differences between Calbinin positive and negative populations. The authors claim a bias of transgenic cFos tagging, both in terms of ratio and expression intensity in the calbindin-positive pyramidal cells in mice exposed to novelty compared to mice remaining in the homecage. There are several flaws in this experiments. First it is underpowered, only including three experimental mice, and the comparison of calb+ cFos+/total cFos+ ratios between homecage and novelty exposed mice makes little sense. It is clear even in this limited sample (figure 4B) that the fraction of calb+ neurons/total cFos + is roughly 50% in the novelty group. Further it is unclear how to interpret the changes observed in GFP intensity in terms of endogenous cFos expression. Finally, these data do not fit the main theme of this technical study.

Overall, while there is some clear technical advance the data generated do not provide much additional insight into hippocampal spatial coding or memory storage.

2. While I appreciate the authors addressing the challenge of disambiguating direct light evoked spikes from spikes generated by presynaptic stimulation it is unclear why they moved to the DG/CA3 circuit. These data would have been stronger if generating in the same circuit (CA3 presynaptic expression of ChR2 instead of the DG/CA3 circuit). Is there a justification why this was not performed?

3. Regarding the cFos labeling experiment- first a larger sample is required. Next. It is clear even in this limited sample (figure 4B) that the fraction of calb+ neurons/total cFos + is roughly 50% in the novelty group. The appropriate analysis should compare Calb+/cFos+ to calb-/cFos+ population adjusted for their relative abundance. Further it is unclear how to interpret the changes observed in GFP intensity in terms of endogenous cFos expression. Is there justification for this measure using this mouse line? This should be discussed.

*Reviewer #3:*

In this paper, the author bring the proof of concept that they can combine juxtacellular single units in freely moving mice with opto-tagging methods. They use this method to show that calbindin-positive neurons are weakly spatially modulated on an o-shaped linear track, but preferentially recruited into spatial memory engrams. The article is clear, concise and well written and is bringing the addition of opto-tagging to the very challenging method of juxtacellular recordings in freely moving animals. The discussion highlights future potential uses of this method. Overall, I do not have major criticisms, but I think the article would benefit from better highlighting the advantages of the technique: opto-tagging in regular extracellular recordings already allows for the recordings of genetically defined subgroups of cells, while "simple" juxtacellular labeling and reconstruction allows for post-hoc characterization of the recorded cellular type. What's the advantage of combining the 2 techniques?

---

## [Author Response]

Essential revisions:While all three reviewers were impressed by the technical tour-de-force presented in this manuscript, they also agreed that the last part of the study regarding cFos expression is underpowered and would be better suited in a future study focusing on this particular question.

We thank the Reviewers for the positive assessment of our work. In line with the Reviewers’ suggestions, we have now removed the cFos expression analysis from the revised manuscript, and focused on the technical aspects of the new technique.

We have removed former Figure 4 and the corresponding parts of the manuscript referring to these data.

1) The main comment is that the authors should reframe their study as a technical report only, focusing on the combination of opto-tagging (including cell type and synaptic interaction) and juxtacellular recording in freely moving animals. In other words, the study should only include the first three figures, and the abstract should be rephrased to only present the technical aspect of the work. The study should then provide an in-depth discussion of the potential advantages and challenges of combining the two techniques (this is only briefly discussed in the present manuscript).

Following the Reviewers’ suggestion, we have reframed the study as a technical report only. We have re-written large parts of the abstract, introduction, results and discussion to address this point. For details, see response to Reviewer 3 below.

2) In the present form of the manuscript, it is not immediately clear how Figure 2 relates to the rest of the study, which won't be the case if this the study is framed as a technical report only. The disambiguation between direct and indirect light-evoked spikes is interesting, but it would have been more natural to carry out such experiments in the CA3/CA1 pathway. Some justification of this choice, if possible, would be helpful to the reader.

We apologize for not having sufficiently clarified the rationale behind our choice for validating out opto-juxtacellular approach in the DG-to-CA3 synapse, instead of the more logical choice of the Schaffer (CA3-to-CA1) collaterals. In the revised manuscript, we clarify this issue. Moreover, to rigorously address this comment, we have performed the additional experiments suggested by the Reviewers. Specifically, we have expressed ChR2 in the CA3, and performed opto-juxtacellular recordings from postsynaptic CA1 neurons. These results are presented in the revised Figure 2A-E. We thank the Reviewers for this suggestion, since the results of these additional experiments greatly strengthens the conclusions of our study. For details, see response to Reviewer 2 below.

3) Firing rate is normalized in figure 3I and 3J – but it would be interesting to know what the max firing rate is for each cell (just add the Hz to each row).

We tried to add the max firing rates next to each raw in Figure 3I,J, but the text becomes too small and difficult to read. As alternative, we now provide max firing rate bar graphs next to the linearized maps in Figure 4I,J. Moreover, we now provide a new Source Data Table (‘Figure 3—SourceData’) which includes mean and peak firing rate values for all cells. For details, see response to Reviewer 1 below.

Reviewer #1:In this manuscript by Ding et al. from the group of Andrea Burgalossi, the authors use a sophisticated combination of juxtacellular recordings and optogenetics to study the spatial firing properties of genetically defined neurons in CA1 of the hippocampus. They first focus on showing that calbindin-positive pyramidal cell were weakly spatially tuned compared to calbindin-negative pyramidal neurons. The authors finish with a brief analysis showing that cFos expressing cells were often calbindin-expressing and this increased with spatial exploration. Overall this is an impressive body of work and will be informative to those interested in heterogeneity in CA1 pyramidal cells, engram recruitment, and more generally this paper established an important technique that can be applied throughout the brain.Overall enthusiasm for the paper is somewhat muted given the finding from the McHugh lab that cfos engram cells are not highly spatial, however the novelty here is the additional link to calbindin expression in addition to the novelty and impressive nature of the methodology used.1. One concern I have is the low number of neurons in the Calbindin positive group (n=7). Of these 7 cells, at least 3 or 4 of them seem as spatial as the calbindin negative group. I realize these are hard experiments, and I don't think more data is needed – but perhaps a more thorough analysis of the data is needed (lap to lap variability, control for firing rates (see below)), etc.

We agree with the Reviewer that our observations are based on a relatively small number of observations (acknowledged now in the revised manuscript, see below).

To rigorously address this point, we have done the following:

1. We have performed additional experiments in freely-moving animals. These additional experiments resulted in larger dataset of neurons included in the spatial analysis (n=38; see Figure 3I-K). The results are fully consistent with the initial dataset, thereby further strengthening the conclusions of our work.

2. We have performed additional analysis. Specifically, we quantified the stability of spatial firing by Pearson’s linear correlation coefficient between rate maps computed for the first and second half of the recording (as e.g. in Agnihotri et al., 2004; Grieves et al., 2020; Preston-Ferrer et al., 2016). We have also confirmed that our conclusions are not significantly biased by inclusion criteria and analysis parameters.

The new data are shown in revised Figure 3I-K and Figure 3—figure supplement 2 and referred to in the revised text (results):

“We found that Calb1-positive neurons conveyed significantly less spatial information (Calb1-positive, 0.65 ± 0.35 bits/spike, n=12; Calb1negative, 1.56 ± 0.79 bits/spike, n=26; p=0.0003) and displayed a higher sparsity index (Calb1-positive, 0.53 ± 0.15, n=12; Calb1-negative, 0.31 ± 0.17, n=26; p=0.0003) (Figure 3K), indicating a more diffuse firing pattern compared to Calb1-negative cells. […] Indeed, in line with their weaker spatial tuning (Figure 3K), spatial firing in Calb1-positive neurons was significantly less stable than in Calb1negative cells (mean correlation coefficient, Calb1-positive, 0.33 ± 0.35; n = 12; Calb1-negative, 0.68 ± 0.31; n = 26; p = 0.0019; see Figure 3—figure supplement 2).”

Moreover, in the methods section we state that:

“Applying more stringent inclusion criteria (e.g. >3 laps, > 100 spikes) and/or different speed thresholds (1-10 cm/s) led to qualitatively similar results and did not change the statistical significance of our findings (p<0.05 for Figures 3I-K)”.

In the revised manuscript (Discussion), we also acknowledge that our conclusions rest on a small number of observations:

“While this finding is in line with previous work (e.g. Danielson et al., 2016; Geiller et al., 2017; Mizuseki et al., 2011; Navas-Olive et al., 2020; Sharif et al., 2021), we acknowledge that it rests on a limited number of observations (Figure 3I-K).”

2. It also appears that the firing rates of calbindin-positive neurons is lower (at least from the example in figure 3F this seems to be the case. I think the authors will need to subsample spiking from the calbindin-negative group to match the low firing rate of the calbindin positive population and then compute information and sparsity (I understand these things so account for firing rate, but a subsampled analysis would help to void this concern)).

We apologize with the Reviewer for not having optimally presented this aspect in the previous version of the manuscript. *Average firing rates* of *Calb1*-positive and *Calb1*-negative neurons are virtually identical (*Calb1*-positive, 2.17 ± 2.02 Hz, n=15; *Calb1*-negative, 2.15 ± 2.10 Hz, n=39; p = 0.86) and this is also true for the subset of neurons included in the spatial analysis (*Calb1*-positive, 2.19 ± 1.79 Hz, n=12; *Calb1*-negative, 2.59 ± 2.28 Hz, n=26; p = 0.88) (see also Author response image 1) Hence, down-sampling of spike rates is not required. However, as correctly pointed out by the Reviewer, *peak firing rates* (on the rate maps) of *Calb1*-negative neurons tend to be higher than (though not significantly different from) that of *Calb1*-positive cells (*Calb1*-positive: 14.2 ± 10.4, n=12; *Calb1*-negative: 8.49 ± 6.10; n=26; p = 0.087). This different trend between average and peak firing rates is indeed consistent with the fact that *Calb1*-negative neurons are more spatially modulated (i.e. similar average activity level, but more spatially-localized, resulting in a trend towards higher peak firing rates).

**Author response image 1. sa2fig1:** Average and peak firing rates of Calb1-positive and Calb1-negative neurons. (A)Box plots showing average firing rates for all Calb1-positive (n=15) and Calb1-negative neurons (n=39) recorded in freely moving animals (B) Average and Peak firing rates for the subset of neurons included in the spatial analysis (n=12 Calb1positive, n=26 Calb1-negative neurons; see Materials and methods).

We now provide max firing rate bar graphs next to the linearized maps in Figure 4I,J. Moreover, we now provide a new a new Source Data Table (‘Figure 3—SourceData’) which includes mean and peak firing rate values for the corresponding cells.

Reviewer #2:The manuscript of Ding et al. is primarily designed as a technical study, describing the synthesis of two challenging approaches that have hitherto have not been attempted in a single preparation. The main experiments describe the bringing together of in vivo juxtacellular recording of mouse hippocampal CA1 pyramidal cells and cell-type specific optoID in behaving mice. Together these approaches allow the generation of ground truth data on in vivo physiology across the levels of anatomy, genetic marker expression, and anatomical location. They demonstrate the feasibility of the approach both in anesthetized and freely moving mice, as well as providing additional data in Figure 2 speaking to the differences in light-evoked spike latency between direct stimulation of ChR2 expression neurons and neurons post-synaptic to stimulated ChR2 expressing axon terminals. These is a nice confirmation on the specificity of the techique although these data would have been stronger if generating in the same circuit (CA3 presynaptic expression of ChR2 instead of the DG/CA3 circuit).The authors validate the approach in the CA1 region of the hippocampus, addressing the previously noted distinction in spatial coding between superficial (calbindin positive) and deep (calbindin negative) CA1 pyramidal cells (see studies by Danielson et al. 2016, Mizuseki et al. 2011, Fernandez Ruiz et al. 2017Valero et al. 2015). The data included reinforce the previous findings they reference, but offer no new insights.1. Finally, a final experiment uses the cFos-tTA mouse to label neurons expressing cFos ('engram neurons ') following exploration of a novel environment to examine differences between Calbinin positive and negative populations. The authors claim a bias of transgenic cFos tagging, both in terms of ratio and expression intensity in the calbindin-positive pyramidal cells in mice exposed to novelty compared to mice remaining in the homecage. There are several flaws in this experiments. First it is underpowered, only including three experimental mice, and the comparison of calb+ cFos+/total cFos+ ratios between homecage and novelty exposed mice makes little sense. It is clear even in this limited sample (figure 4B) that the fraction of calb+ neurons/total cFos + is roughly 50% in the novelty group. Further it is unclear how to interpret the changes observed in GFP intensity in terms of endogenous cFos expression. Finally, these data do not fit the main theme of this technical study.Overall, while there is some clear technical advance the data generated do not provide much additional insight into hippocampal spatial coding or memory storage.

We thank the Reviewer for these constructive suggestions on how to improve our cFos expression data. In line with the comments from the other Reviewers, we opted for removing these experiments in the revised manuscript. We agree that these experiments would be better suited in a future study focusing on this particular question.

Former Figure 4 has been removed, along with the corresponding sections in the Results and Discussion.

2. While I appreciate the authors addressing the challenge of disambiguating direct light evoked spikes from spikes generated by presynaptic stimulation it is unclear why they moved to the DG/CA3 circuit. These data would have been stronger if generating in the same circuit (CA3 presynaptic expression of ChR2 instead of the DG/CA3 circuit). Is there a justification why this was not performed?

We agree with this Reviewer that in the previous version of the manuscript, it was not immediately evident why we chose to validate our opto-tagging approach in the DG-to-CA3 synapses – instead of the more logical choice of the Schaffer collaterals. In the revised manuscript we have done the following:

1. We clarified the rationale behind our choice. Specifically, had initially opted for the dentate mossy fiber terminals, since they are known to be powerful and reliable (‘detonator’- synapses) (e.g. Henze et al., 2002; Neubrandt et al., 2018; Vyleta et al., 2016) unlike most other hippocampal (and cortical) synapses. Given the robustness and reliability of this synapse, we thought this could serve as the ‘most stringent test’ for validating our latency-based criteria for discriminating directly versus indirectly activated neurons.

2. To rigorously address this point, we have performed the additional experiments suggested by the Reviewer. Specifically, we have injected AAV-CAG-ChR2mCherry in the CA3, and performed opto-juxtacellular recordings from postsynaptic CA1 pyramidal neurons. This new dataset (n=11 cells) further confirms our latency-based criteria, in that synapticallyactivated CA1 neurons have significantly longer latencies compared to directly light-activated ones.

The new dataset of synaptically-activated CA1 neurons is shown in revised Figure 2A-E, and referred to in the Results section:

“(…) we injected AAV-CAG-ChR2-mCherry in the CA3 of wild-type mice (Figure 2A). By these means, we obtained strong labelling of CA3 axon terminals (Schaffer collaterals), extending into the CA1 strata radiatum and oriens (Figure 2B). […] At the population level, the response latencies of indirectly light-activated CA1 neurons were significantly longer than those of directly lightactivated CA1 Calb1-positive cells (direct light-activation, mean latency = 2.7 ± 0.3ms, n=10; indirect light activation, mean latency = 7.0 ± 1.8 ms, n=11； p=0.0010; Figure 2J), thereby further validating our latency-based criteria for cell type classification.”

To clarify the rationale behind our choice of the dentate mossy fibers, we state the following (results):

“The above data indicate synaptically-activated neurons can be reliably discriminated from directly light-activated cells, at least when ‘weak and unreliable’ synapses – like typical cortical synapses, including Schaffer terminals (Csicsvari et al., 2000; Sayer et al., 1990, 1989) – are activated. […] To this end, we performed a subset of experiments on the hippocampal mossy fiber synapse, which is among the strongest and most reliable connections in the nervous system (Henze et al., 2002; Neubrandt et al., 2018; Vyleta et al., 2016)..”

3. Regarding the cFos labeling experiment- first a larger sample is required. Next. It is clear even in this limited sample (figure 4B) that the fraction of calb+ neurons/total cFos + is roughly 50% in the novelty group. The appropriate analysis should compare Calb+/cFos+ to calb-/cFos+ population adjusted for their relative abundance. Further it is unclear how to interpret the changes observed in GFP intensity in terms of endogenous cFos expression. Is there justification for this measure using this mouse line? This should be discussed.

We thank the Reviewer for these constructive suggestions on how to improve our cFos expression analysis. As noted above, we have removed these experiments in the revised manuscript, and focused on the technical aspects of our method.

Reviewer #3:In this paper, the author bring the proof of concept that they can combine juxtacellular single units in freely moving mice with opto-tagging methods. They use this method to show that calbindin-positive neurons are weakly spatially modulated on an o-shaped linear track, but preferentially recruited into spatial memory engrams. The article is clear, concise and well written and is bringing the addition of opto-tagging to the very challenging method of juxtacellular recordings in freely moving animals. The discussion highlights future potential uses of this method. Overall, I do not have major criticisms, but I think the article would benefit from better highlighting the advantages of the technique : opto-tagging in regular extracellular recordings already allows for the recordings of genetically defined subgroups of cells, while "simple" juxtacellular labeling and reconstruction allows for post-hoc characterization of the recorded cellular type. What's the advantage of combining the 2 techniques?

We thank the Reviewer for the positive assessment of our work. Following this suggestion, we have now rewritten large parts of the abstract, introduction and discussion, and focused on key methodological aspects, advantages/disadvantages of our opto-juxtacellular technique. For details, see the revised text.

We have significantly changed the abstract, introduction and discussion (additional significant changes are copy/pasted here below) to address the technical aspects of the technique, including advantages and disadvantages.

Regarding the advantages of combining juxtacellular and optogenetic techniques, in the Discussion we state we state that:

“(our method) enables targeting of juxtacellular recordings to a genetically pre-defined cell class; hence, it represents a significant advance over current juxtacellular protocols, where individual neurons are blindly sampled within a target structure (Averkin et al., 2016; Diamantaki et al., 2018; Katona et al., 2014; Lapray et al., 2012; Tang et al., 2014b; Valero et al., 2015)”.

As additional advantages, we state in the discussion that:

“Our combined optojuxtacellular method also provides several advantages over ‘conventional’ extracellular opto-tagging approaches. […] Third, post-hoc morphological and molecular expression analysis of the recorded/labelled cells provides access to structural features that are typically not accessible with alternative techniques, thereby enabling unequivocal cell type classification.”.

For disadvantages, we state in the Discussion that:

“While it is technically possible to perform multiple opto-juxtacellular recordings within individual experiments (Figure 3—figure supplement 1), our method still remains laborious and of limited output compared to alternative techniques (e.g. extracellular recordings). […] We envision that opto-juxtacellular datasets – although limited – could still provide a valuable complement to current cell-type classification approaches, for example by offering the possibility of building a classifier, which could then be used for assigning cell identity to ‘blind’ extracellular units (as in e.g. GoodSmith et al., 2017; Tang et al., 2014a)”.

Also related to a technical aspect of the optogenetic approach, we have now quantified the degree of ChR2/Calb1 colocalization via confocal analysis (methods):

“In Calb1-Cre mice injected within the CA1 pyramidal layer, ChR2 was almost exclusively expressed in Calb1-positive neurons (0 out of 272 neurons were ChR2-positive / Calb1-negative), thus confirming the specificity of the Calb1Cre driver line (see also Daigle et al., 2018; Nigro et al., 2018). Viral titers and injection volumes were optimized to obtain high infection efficiency and ChR2 expression levels (~95.8% of Calb1-positive neurons were also ChR2-positive; 261/272 neurons), thereby minimizing the occurrence of false-negatives in our opto-tagging experiments.”